# Targeting the MEK/ERK Pathway to Suppress P-Glycoprotein and Reverse Carfilzomib Resistance in Multiple Myeloma

**DOI:** 10.3390/ijms262311448

**Published:** 2025-11-26

**Authors:** Lidia A. Laletina, Anastasiia I. Cherkasova, Ekaterina A. Scherbakova, Pavel S. Iamshchikov, Natalia A. Koroleva, Anna A. Lushnikova, Alexey A. Komissarov, Nikolay Kalitin, Natalia I. Moiseeva

**Affiliations:** 1N.N. Blokhin National Medical Research Center of Oncology, Kashirskoe Shosse 24, 115478 Moscow, Russia; panlidia@gmail.com (L.A.L.); a-cherkasova2000@mail.ru (A.I.C.); scherbakovakatya108@gmail.com (E.A.S.); nat.korole@yandex.ru (N.A.K.); lan21@yandex.ru (A.A.L.); f.oskolov@mail.ru (N.K.); 2Center for Systems Bioinformatics, Tomsk National Research Medical Center, Russian Academy of Sciences, 634050 Tomsk, Russia; pavel.iamshchikov@tnimc.ru; 3I.V. Davydovsky Moscow City Clinical Hospital, Moscow Department of Healthcare, 117463 Moscow, Russia; komissarovlexa@yandex.ru; 4Laboratory of Atherothrombosis, The Russian University of Medicine, 127473 Moscow, Russia

**Keywords:** P-glycoprotein, carfilzomib, multiple myeloma, ulixertinib, cobimetinib, MAPK pathway, MEK inhibitors, ERK inhibitors, RNAseq

## Abstract

Carfilzomib (CFZ) is a cornerstone in the treatment of relapsed multiple myeloma (MM). However, its efficacy is limited by resistance mediated by the overexpression of the ABC-transporter P-glycoprotein (P-gp). The signaling pathways driving the emergence of P-gp in MM remain unclear. To investigate this, we generated CFZ-resistant AMO-1/CFZ cells with P-gp overexpression by long-term selection. RNA sequencing of control AMO-1 and AMO-1/CFZ, sorted into two subpopulations, P-gp HIGH and P-gp LOW, implicated the Ras/MEK/ERK pathway as the most likely signaling cascade involved in P-gp upregulation. We therefore evaluated two clinically used MAPK pathway inhibitors, cobimetinib and ulixertinib, for their ability to re-sensitize AMO-1/CFZ cells to CFZ. Co-administration at non-toxic concentrations enhanced sensitivity 5-fold with cobimetinib and 17-fold with ulixertinib. Analysis of the combined MTT assay results, rhodamine efflux experiments, molecular docking, and Western blotting revealed distinct actions. Ulixertinib primarily functions as a potent direct P-gp inhibitor. Conversely, non-toxic concentrations of cobimetinib sensitizes cells by suppressing MAPK signaling, though it also exhibits P-gp inhibition at higher concentrations. At the IC_50_ concentration, both inhibitors reduced P-gp expression. In conclusion, combining CFZ with MAPK pathway inhibitors like cobimetinib or ulixertinib represents a promising strategy to overcome P-gp-mediated resistance in MM.

## 1. Introduction

Multiple myeloma (MM) is a heterogeneous hematological malignancy characterized by the clonal proliferation of plasma cells in the bone marrow [1]. Significant progress in MM treatment has been achieved with the introduction of proteasome inhibitors (PIs), such as carfilzomib (CFZ), which has become a cornerstone in the therapy of patients with relapsed disease [2]. Despite its clinical efficacy, the development of acquired resistance remains a major obstacle to long-term disease control. One of the key and well-characterized mechanisms of resistance to chemotherapeutic agents is the increased activity of transmembrane drug efflux transporters, primarily P-glycoprotein (P-gp, MDR1, ABCB1). While CFZ demonstrates sustained activity in the context of resistance to bortezomib (BTZ), a first-generation PI [3,4], evidence suggests that it is not devoid of vulnerability to efflux mechanisms.

P-gp overexpression, often induced by prior therapy, can actively pump CFZ out of tumor cells, reducing its intracellular concentration to subtherapeutic levels and, consequently, minimizing proteasome inhibition and the apoptotic response. This is supported by studies demonstrating that CFZ-resistant MM cell lines exhibit upregulated *ABCB1* gene expression and functional P-gp activity. Burbage et al. used a genetic construct for the controlled overexpression of the *ABCB1* gene in MM cells, which allowed them to study the specific effect of P-gp while minimizing other adaptive resistance mechanisms. Their work provides compelling evidence that CFZ, unlike other PIs, is a substrate for P-gp [5]. A large study by Besse et al. strongly argues that P-gp overexpression and functional activity are a clinically relevant mechanism of resistance to CFZ. In this work, increased *ABCB1* gene expression was detected in cells from patients with plasma cell leukemia—the most aggressive stage of MM [6]. The same study also demonstrated the potent inhibitory activity of nelfinavir and lopinavir (antiviral drugs—HIV protease inhibitors) against P-gp in vitro. In combination with CFZ, these drugs fully restored the sensitivity of resistant cells to therapy, significantly enhancing apoptosis [6]. Thus, P-gp inhibition could be clinically significant for some MM patients for whom CFZ treatment is planned or who are undergoing CFZ therapy.

Attempts to overcome multidrug resistance (MDR) have led to the development of three generations of P-gp inhibitors. The first (verapamil, cyclosporine A) and second (valspodar, biricodar) generations of P-gp inhibitors suffered from either low specificity and serious side effects, or unfavorable pharmacokinetic interactions [7,8,9,10]. Third-generation inhibitors, including zosuquidar, tariquidar, and elacridar (ELA), were developed as highly effective, specific P-gp inhibitors; however, they failed to demonstrate efficacy in clinical trials and have not been approved by regulatory agencies such as the FDA or EMA for widespread use in oncology [11,12]. The scientific community is now developing several advanced strategies, the combination of which may effectively overcome MDR: developing a new generation of P-gp inhibitors [13,14,15,16], utilizing innovative drug delivery systems, and implementing principles of personalized medicine [17]. A promising approach under consideration is drug repurposing—using non-oncological/oncological drugs for other indications as P-gp inhibitors (e.g., imatinib, lopinavir/nelfinavir) [6,18,19]. In this context, the use of low-molecular-weight kinase inhibitors is frequently considered [20,21,22]. However, their selection requires knowledge of the activation of specific signaling pathways leading to P-gp upregulation.

A key regulatory component of the *ABCB1* gene involves intracellular signaling pathways such as PI3K/Akt/mTOR, MAPK (Mitogen-Activated Protein Kinases), PKC (Protein Kinase C), NF-κB (Nuclear Factor kap-pa-light-chain-enhancer of activated B cells), Nrf2 (Nuclear factor erythroid 2–related factor 2), Hedgehog (Hh), Wnt/β-catenin, Notch, and Hippo [23,24,25], which integrate diverse extracellular and intracellular stimuli. These pathways frequently converge on transcription factors AP-1, NF-κB, C/EBPβ, and c-ETS to directly drive *ABCB1* expression and promote P-gp-mediated therapy resistance [26,27,28,29]. The specific pathway activated is determined by the initial signal and cellular context.

For instance, in a subpopulation of leukemia stem cells from drug-resistant K562/ADM, a significant increase in P-gp and BCRP expression was observed due to the aberrant activation of the NF-κB pathway. Parthenolide, a specific NF-κB inhibitor, sensitized the cells and reduced P-gp expression [30]. Acquired lenvatinib resistance in hepatocellular carcinoma was driven by an EGFR/STAT3 axis, where STAT3 directly bound the *ABCB1* promoter to upregulate P-gp and enhance drug efflux, and inhibiting either EGFR or STAT3 restored drug sensitivity [31]. Zhang S. and Wang Y. investigated the effect of the natural compound deoxyshikonin on cisplatin resistance in non-small cell lung cancer cells. They established that the resistance mechanism was associated with hyperactivation of the PI3K/Akt signaling pathway, which mediated the increased expression and functional activity of P-gp. Deoxyshikonin effectively suppressed Akt phosphorylation, leading to a significant reduction in *ABCB1* mRNA and protein levels [24].

However, there is a scarcity of direct evidence demonstrating that CFZ activates specific signaling pathways leading to the direct upregulation of *ABCB1* in MM cells. The study by Riz et al. provides a comprehensive investigation explaining how CFZ can indirectly, through activation of the non-canonical p62-Nrf2 pathway, lead to increased *ABCB1*/P-gp expression [32]. Therefore, identifying the specific signaling pathways that regulate *ABCB1*/P-gp expression in MM cells remains a relevant and pressing scientific challenge.

Thus, despite the emergence of novel MM therapies such as CAR-T cells and bispecific antibodies [33,34,35], CFZ, a P-gp substrate, remains a core component of treatment for patients with relapsed MM. Therefore, efforts continue to develop effective methods to suppress P-gp function in MM cells. In our study, we focus on identifying the signaling pathways and specific targets (kinases) that lead to increased expression and activation of P-gp in MM cells. These findings may form the basis for developing a combination therapy of CFZ with selective kinase inhibitors to overcome drug resistance.

## 2. Results

### 2.1. Development and Characterization of a Carfilzomib-Resistant Cell Line

The AMO-1/CFZ resistant cell line was established by long-term exposure of the MM AMO-1 cell to escalating doses of CFZ. AMO-1/CFZ cells are 25 times more resistant to CFZ than the AMO-1 cells (IC_50_ = 11.4 ± 3.0 and IC_50_ = 295.5 ± 57.2 for AMO-1/CFZ and AMO-1 cells, respectively). Cell survival graphs (a) and the statistical significance of the results (b) are presented in Figure 1. Also, the expression of P-gp in both cell lines was analyzed at the mRNA level by quantitative real-time PCR (RT-qPCR) and protein level by Western blot. Expression of the *ABCB1* gene, which encodes P-gp, was 996.5 times higher in AMO-1/CFZ cells than in the parental AMO-1 cells (Figure 1c). P-gp protein was undetectable in AMO-1 cells, whereas in AMO-1/CFZ its expression was significantly increased (Figure 1d). To confirm that P-gp overexpression is the primary factor determining AMO-1/CFZ cells’ resistance to CFZ, combined MTT assays were performed. The cells were treated with CFZ alone or in combination with a non-toxic concentration of ELA, a P-gp inhibitor. The IC_50_ values were as follows: 34.0 ± 10.1 nM and 24.9 ± 2.9 nM in AMO-1 cells, 801.2 ± 156.2 nM and 27.9 ± 9.5 nM in AMO-1/CFZ (for CFZ and CFZ + ELA treatment, respectively). Overall, P-gp inhibition by ELA reduced the IC_50_ of CFZ in the resistant cells by 28.7-fold, restoring sensitivity to a level comparable to the AMO-1 cells. Dose–response curves were plotted based on the results (Figure 1e), and the statistical significance of the data was determined (Figure 1f).

Thus, the development of CFZ resistance in AMO-1 cells is characterized by a multi-fold increase in P-gp expression. The subsequent addition of a P-gp inhibitor completely re-sensitized the AMO-1/CFZ cells, restoring their sensitivity to the level of the parental line. These results demonstrate the primary role of P-gp in conferring CFZ resistance in this model.

### 2.2. Isolation of P-gp HIGH and P-gp LOW Subpopulations from AMO-1/CFZ Cells for RNA-Seq Analysis

The next stage of our work involved searching for signaling pathways that lead to the activation of P-gp expression using RNA sequencing (RNA-Seq). To make this search more accurate, we used flow cytometry to determine whether the entire AMO-1/CFZ cell population expresses P-gp on the cell surface. First of all, no P-gp expression was detected in AMO-1 cells (Figure 2), which is consistent with the Western blot analysis results presented above (Figure 1d). A pronounced P-gp expression was detected in 65.4 ± 10.2% of AMO-1/CFZ cells (Figure 2). Notably, P-gp expression was unimodal across the population; there was no clear bimodal distinction into P-gp(–) and P-gp(+) subpopulations. This was reflected in the dot blot and fluorescence intensity histogram, not as a division of populations with a double-humped peak, but rather as a shift in the cell cloud along the signal intensity scale. Based on these results, we decided to use cell sorting to divide the AMO-1/CFZ line into two subpopulations: high P-gp expression (HIGH) and low P-gp expression (LOW).

To achieve a better separation of these populations, gating was performed to ensure that cells with an average signal were not captured during sorting and that only cells with a strong or minimal signal were selected. After each separation into populations, we performed a post-sort analysis to confirm the quality of the separation. This analysis showed that the P-gp LOW population was isolated with 98–99% purity and the P-gp HIGH population with 87–92% purity (Figure 2).

### 2.3. RNA-Seq Identifies Signaling Pathways Activating ABCB1 Expression

To identify signaling pathways associated with P-gp overexpression, the AMO-1/CFZ was separated by cell sorting into AMO-1/CFZ_P-gp_LOW and AMO-1/CFZ_P-gp_HIGH subpopulations. A total lysate from the native AMO-1 cells was used as a control (CTRL). In total, 12 samples (4 biological replicates per group) were subjected to RNA-Seq.

Principal component analysis (PCA) and hierarchical clustering of the most variable genes demonstrated a clear separation between the CTRL group and the combined AMO-1/CFZ groups (Figure 3b). The CTRL samples formed a distinct cluster, whereas the AMO-1/CFZ_P-gp_HIGH and AMO-1/CFZ_P-gp_LOW samples showed significant overlap on the PCA plot and did not segregate upon clustering (Figure 3a), indicating high transcriptomic similarity between these two subpopulations.

Differential gene expression (DGE) analysis confirmed these observations. Comparisons of AMO-1/CFZ_P-gp_HIGH vs. CTRL and AMO-1/CFZ_P-gp_LOW vs. CTRL revealed 6125 and 6033 significantly differentially expressed genes (FDR < 0.05), respectively. In contrast, a direct comparison between the AMO-1/CFZ subpopulations identified only 24 differentially expressed genes. Among these, the *ABCB1* gene, which encodes P-gp, was among the most significantly upregulated in the AMO-1/CFZ_P-gp_HIGH group, confirming the efficacy of the cell sorting (Figure 3c–e). These data suggest that the P-gp_LOW subpopulation has already undergone key transcriptomic changes necessary for drug resistance, while the additional expression of P-gp may be regulated at the post-translational level.

Due to the weak transcriptomic differences between the AMO-1/CFZ subpopulations, signaling pathways for further analysis were selected based on their significant activation in the AMO-1/CFZ_P-gp_HIGH group compared to CTRL (NES > 1.4) and the inclusion of the *ABCB1* gene. The most promising pathways (Figure 3f) in this regard were UV_RESPONSE_UP (NES = 1.85) and KRAS_SIGNALING_UP (NES = 1.4). The UV_RESPONSE_UP pathway represents a complex network activated in response to DNA damage. It is known that UV radiation can directly activate growth factor receptors, such as EGFR, triggering downstream signaling cascades independently of ligands [36]. The MAP kinase cascade plays a central role in this response, involving the sequential activation of kinases that regulate the cell cycle, apoptosis, and transcription factor activity [37]. The KRAS_SIGNALING_UP pathway plays a key role in the pathogenesis of hematologic malignancies, including multiple myeloma, and approved low-molecular-weight inhibitors targeting its components are available. The combination of the latter factors led us to begin investigating the role of KRAS_SIGNALING_UP in P-gp activation in AMO-1/CFZ cells.

### 2.4. Confirmation of the RNA-Sequencing Results Using RT-qPCR

The sequencing results were confirmed by RT-qPCR. Based on the sequencing data, we selected three genes with increased expression (*ABCB1, EPHB2*, and *CROT*) and one gene with decreased expression, *IL6R,* in AMO-1/CFZ P-gp HIGH cells compared to AMO-1 cells. These genes were among the most differentially expressed, and several (*ABCB1, EPHB2, CROT*) are involved in KRAS signaling pathways. It has been verified that the sequencing data are reliable and that these genes exhibit differential expression in resistant and sensitive cells (Figure 4a–d). Based on the results of RT-qPCR, it was determined that in AMO-1/CFZ P-gp HIGH cells, the relative expression of the *ABCB1*, *EPHB2*, and *CROT* genes was increased by 737, 11, and 10 times, respectively, and the expression of the *IL6R* gene was decreased by 3.5 times compared to AMO-1 cells. Also, in AMO-1/CFZ P-gp LOW cells, the relative expression of the *ABCB1*, *EPHB2*, and *CROT* genes was increased by 265.5, 2, and 6.8 times, respectively, and the expression of the *IL6R* gene was decreased by 4.6 times compared to AMO-1 cells.

Furthermore, the 2.7-fold higher *ABCB1* expression in the P-gp HIGH group compared to the P-gp LOW group confirms the successful separation of the subpopulations by cell sorting. Notably, while *CROT* and *ABCB1* were elevated in both resistant subpopulations, *EPHB2* expression was elevated exclusively in the P-gp HIGH cells, suggesting its potential specific role in P-gp-mediated CFZ resistance.

### 2.5. Inhibitor Selection and Cytotoxicity Profiling

To study the effect of the KRAS signaling pathway on P-gp overexpression, it was decided to inhibit this pathway in cells using the already known inhibitors cobimetinib and ulixertinib and evaluate the sensitivity of cells to CFZ, as well as the level of P-gp expression. Cobimetinib is an oral, highly selective allosteric inhibitor of MEK1/2 kinase, while ulixertinib is a selective inhibitor of mitogen-activated protein kinases ERK1 and ERK2.

We first evaluated the cytotoxicity of ulixertinib and cobimetinib in AMO-1 and AMO-1/CFZ cell lines using the MTT assay. For the cobimetinib IC_50_ were 12.4 ± 4.0 and 14.1 ± 2.0 μM, and for the ulixertinib IC_50_ were 31.5 ± 0.5 and 33.4 ± 1.9 μM (for AMO-1 and AMO-1/CFZ cells, respectively). For further studies, low-toxicity doses of 1 μM for cobimetinib and 10 μM for ulixertinib were selected. Cell viability at low-toxicity doses is shown in Appendix A. Cobimetinib exhibited greater toxicity on MM cells than ulixertinib. Cytotoxicity curves for both compounds are shown in Figure 5a,b.

We next investigated whether non-toxic concentrations of cobimetinib and ulixertinib could sensitize AMO-1 and AMO-1/CFZ cells to CFZ. The combination with cobimetinib reduced the IC_50_ of CFZ by 5.6-fold in AMO-1 cells and 3.3-fold in AMO-1/CFZ cells. Ulixertinib was more effective in the resistant line, reducing the CFZ IC_50_ by 17.9-fold in AMO-1/CFZ cells compared to 1.9-fold in the parental AMO-1 cells. Thus, the inhibitors cobimetinib and ulixertinib sensitize cells to CFZ, reducing its IC_50_ by 1.9–17.9 times. Interestingly, cobimetinib more effectively sensitized the treatment-parental AMO-1 cells, whereas ulixertinib was markedly more potent at reversing resistance in the CFZ-resistant AMO-1/CFZ cells. The obtained data and their statistical analysis are presented in Figure 5c–f. The obtained IC_50_ values are presented in Table 1.

### 2.6. Analysis of Gene Expression Following Inhibitor Treatment

We next focused on how cobimetinib and ulixertinib inhibitors affect the expression of some of the most different genes in AMO-1/CFZ P-gp HIGH cells compared to AMO-1, and also wanted to understand whether cobimetinib and ulixertinib affect the expression of genes in the KRAS signaling pathway, of which they are inhibitors. Using RT-qPCR, we analyzed the relative expression of *ABCB1* and *EPHB2*—genes encoding proteins involved in KRAS signaling—in AMO-1 and AMO-1/CFZ cells after 72 h and 96 h exposure to the IC_50_ of each inhibitor (cobimetinib: 12 µM for AMO-1, 14 µM for AMO-1/CFZ; ulixertinib: 32 µM for AMO-1, 36 µM for AMO-1/CFZ). Also, the dependence of the expression of the listed genes on the time of exposure to IC_50_ of these inhibitors on AMO-1 and AMO-1/CFZ cells was determined (Figure 6).

Based on the RT-qPCR results, it can be concluded that expression of the *ABCB1* gene is reduced by cobimetinib in both resistant and naive cells. Cobimetinib suppressed *ABCB1* expression in both cell lines in a time-dependent manner. In AMO-1 cells, it caused a 2.7-fold decrease after 72 h and a 3.6-fold decrease after 96 h. In AMO-1/CFZ cells, it induced a consistent 1.6-fold decrease at both time points. At the same time, ulixertinib does not significantly reduce *ABCB1* gene expression in any of the cell lines, indicating a different mechanism of its synergistic action with CFZ. *EPHB2* gene expression is increased by both ulixertinib and cobimetinib in a time-dependent manner in AMO-1 cells, with a significant increase in expression at 96 h, and is also increased by these inhibitors in resistant AMO-1/CFZ cells. The *EPHB2* gene encodes a receptor tyrosine kinase whose expression is linked to MAPK pathway activity and is feedback-upregulated upon pathway inhibition [38]. The effects of cobimetinib and ulixertinib on the expression of two other genes used to validate the sequencing are shown in Appendix A.

### 2.7. Functional Assessment of Inhibitor Binding to P-Glycoprotein Using a Rhodamine 123 Efflux Assay

To investigate potential interactions between P-gp and CFZ, cobimetinib, or ulixertinib, we employed the Rhodamine 123 (Rh123) efflux assay. Initial loading of AMO-1/CFZ cells with Rh123 resulted in 97.8% of cells being stained. Subsequent incubation in drug-free medium led to efficient Rh123 efflux, with only 13.9% of cells retaining the dye, confirming robust P-gp activity. As expected, the reference P-gp inhibitor ELA significantly inhibited Rh123 efflux at 1 µM and 10 µM (97.8% and 99.1% of cells stained, respectively; Figure 7a). CFZ exhibited a concentration-dependent effect: at 0.5 µM, it did not inhibit efflux (8.1% cells stained), but at 20 µM, it competed with Rh123, reducing efflux (28.9% cells stained), identifying it as a P-gp substrate (Figure 7b). Cobimetinib and ulixertinib also showed concentration-dependent inhibition. At 1 µM, cobimetinib did not affect efflux (10.5% cells stained), but at 20 µM, it potently inhibited P-gp (87.1% cells stained) (Figure 7c). Similarly, ulixertinib had no effect at 1 µM (9.5% cells stained) but strongly blocked efflux at 10 µM and 20 µM (87.6% and 92.5% cells stained, respectively), with its efficacy at 20 µM being comparable to ELA (Appendix A) (Figure 7d). These results indicate that at high concentrations, both cobimetinib and ulixertinib act as P-gp inhibitors. This is consistent with our MTT assay data, which showed a 25-fold reduction in CFZ sensitivity in P-gp-high AMO-1/CFZ cells, while sensitivity to cobimetinib and ulixertinib remained unchanged between the cell lines. The differential effect—CFZ as a substrate and the inhibitors as potential blockers—explains the synergy observed in combination treatments.

### 2.8. Molecular Docking of P-Glycoprotein and Cobimetinib and Ulixertinib

Sensitization and Rh123 efflux assays indicated that ulixertinib and cobimetinib may function as P-gp inhibitors. To further investigate this, we performed molecular docking for all molecules of interest, including CFZ, as well as several known P-gp substrates and inhibitors, and two other PIs, BTZ and ixazomib (IXZ).

The docking results indicated that CFZ had a higher binding score than other known P-gp substrates, such as paclitaxel and doxorubicin (−9.627 vs. −8.226 and −7.483, respectively). The model predicted three Pi-pi stacking interactions and one hydrogen bond (H-bond) for CFZ (Figure 8a,b). The binding scores for cobimetinib (Figure 8c,d) and ulixertinib Figure 8e,f) were lower than that of the potent inhibitor ELA (−8.783 and −7.698 vs. −9.852, respectively), which correlates with the results of the Rh123 efflux experiments. Docking scores, binding positions, and interaction details are provided in Table 2 and Appendix A.

The docking simulations visually demonstrated that all tested substrates occupy a central position within the P-gp (Appendix A). In contrast, known inhibitors, BTZ and IXZ, are arranged radially in the molecule (Appendix A). Furthermore, substrate binding was characterized by interaction with the GLU875 residue, while all inhibitors interacted with PHE770.

To model competitive inhibition, we performed a double-docking procedure using a two-step algorithm. First, ELA and the putative inhibitor cobimetinib, or ulixertinib, were docked (Appendix A). The protein-inhibitor complex was then exported, and a second docking was performed with CFZ. For combinations of CFZ with ELA and cobimetinib, the number of bonds formed by CFZ was significantly reduced, potentially preventing it from occupying the position necessary for efficient efflux by the P-gp. Although the co-docking with ulixertinib did not show a significant change in the number of bonds, the character of these interactions was altered, with new potential bonds forming with other amino acids.

### 2.9. Western Blot Analysis of Signaling Pathways After Inhibitor Exposure

In the final stage of the study, we analyzed the activity of the MAPK signaling pathway in AMO-1 and AMO-1/CFZ cells in the presence of cobimetinib and ulixertinib, in order to ascertain the impact of sensitizing these cell cultures to CFZ. We also aimed to verify whether suppressing MAPK signaling results in a reduction in P-gp expression at the protein level in AMO-1/CFZ cells. For these purposes, we opted for a 72 h incubation period with inhibitors, which aligns with our sensitization experiments and is sufficient to detect any changes in P-gp protein levels. Two concentrations were used for cobimetinib and ulixertinib: a low-toxic concentration of 1 μM and 10 μM, respectively, and an IC_50_ concentration of 14 μM and 32/36 μM (Figure 9).

The effect of cobimetinib on AMO-1 (Figure 9a,c) and AMO-1/CFZ (Figure 9a,e) cells resulted in a more than 10-fold increase in p-MEK1/2 at both non-toxic concentrations and at the IC_50_ concentration. Overall, treatment with both inhibitors led to a significant accumulation of p-MEK1/2 kinase in almost all cases, which is consistent with literature data showing a decrease in p-MEK1/2 at the earliest exposure stages, from several minutes to two hours, followed by a time-dependent increase in p-MEK1/2, probably due to the feedback principle [39,40]. We continue to detect this increase 72 h after exposure. At the same time, a significant decrease in the amount of downstream p-ERK1/2 kinase was observed under the influence of cobimetinib, by 4–5 times, except when 1 μM was used on AMO-1, which indicates stable disruption of this signaling pathway. AMO-1/CFZ cells exposed to cobimetinib at an IC_50_ concentration showed a significant twofold decrease in P-gp (Figure 9a,b). Overall, the effect of cobimetinib on AMO-1 and AMO-1/CFZ cells was similar.

The effect of ulixertinib on AMO-1 (Figure 9a,d) and AMO-1/CFZ (Figure 9a,f) cells was different. In AMO-1 cells, a significant increase in p-MEK1/2 was observed; the level of p-ERK1/2 did not differ from the control level and appeared to have returned to its initial state. However, the downstream kinase p-p90RSK was suppressed by two- to threefold, indicating that the pathway was not functionally active after 72 h. After 72 h of ulixertinib exposure, no significant differences in MAPK pathway activity were observed in AMO-1/CFZ cells, although a tendency towards increased p-MEK1/2 and decreased p-p90RSK was noted. At the IC_50_ concentration, however, ulixertinib reduced P-gp expression by an average of 30% (Figure 9a,b), albeit slightly weaker than under the influence of cobimetinib.

The amount of c-Raf kinase level remained unchanged in any of the exposure variants. It is worth noting that, although we evaluated P-gp expression in AMO-1 cells as they may respond to stress exposure by increasing their expression, we did not detect such an effect when using cobimetinib and ulixertinib.

Thus, the use of high concentrations of both cobimetinib and ulixertinib led to a decrease in P-gp expression, indicating the involvement of the MAPK pathway in its activation in MM cells.

## 3. Discussion

Our data strongly suggest that P-gp plays a determining role in the development of resistance to CFZ in the model used. Key evidence for this includes the significant P-gp overexpression in resistant AMO-1/CFZ cells, coupled with the complete restoration of CFZ sensitivity upon its inhibition by ELA. This is further supported by Rh123 efflux assays and docking experiments demonstrating that CFZ is a P-gp substrate (Figure 1, Figure 2, Figure 7b and Figure 8a,b). This finding is consistent with the existing scientific literature. The work of Besse et al. [6] also provides compelling evidence that P-gp overexpression is a significant mechanism of resistance specifically to CFZ, and not only to classical chemotherapeutic agents. In that study, significant overexpression of the *ABCB1* gene and P-gp protein was identified in generated resistant MM cell clones, accompanied by high functional activity of this transporter, as confirmed by reduced accumulation of the fluorescent substrate Mitotracker Green FM. While the role of P-gp in resistance to CFZ is becoming evident, the key question of which specific signaling pathways mediate the induction of its expression in response to treatment remains open.

While the progression and therapy resistance of MM are known to involve the hyperactivation of key pathways such as PI3K/AKT/mTOR, RAS/MAPK, and NF-κB [41], the specific cascades crucial for P-gp upregulation in response to CFZ remain elusive. To systematically identify which of these pathways are implicated in P-gp activation, we performed RNA sequencing to obtain a comprehensive profile of the transcriptomic changes following CFZ exposure.

Based on RNA-seq analysis of three sample groups (Figure 3), we selected the RAS/RAF/MEK/ERK signaling cascade, which demonstrated significant activation in AMO-1/CFZ cells compared to the sensitive AMO-1 line (NES > 1.4). The RAS/RAF/MEK/ERK pathway represents one of the key signaling cascades that regulates *ABCB1* expression at the transcriptional level, frequently through transcription factors such as Ets-1 [42,43]. This study aimed not only to identify signaling pathways that induce P-gp expression but also to evaluate their inhibitors as means to overcome resistance. Targeting the RAS-RAF-MEK-ERK pathway represents a promising direction. Approved BRAF inhibitors already exist (e.g., vemurafenib), which is relevant for 5–10% of MM patients with the aggressive *BRAF* V600E-mutant form of the disease and opens possibilities for personalized combination strategies [44]. Despite the high initial efficacy of BRAF inhibitors (vemurafenib, dabrafenib) in *BRAF* V600E-mutant tumors, monotherapy often leads to rapid development of resistance driven by MAPK pathway reactivation [45,46,47]. To overcome this limitation, a strategy of vertical pathway blockade using combinations of BRAF and MEK inhibitors has been proposed. An alternative or complementary approach involves direct ERK inhibition, which potentially enables suppression of the signaling regardless of the reactivation mechanism upstream in the pathway [48,49]. Guided by these principles, we selected the small molecule inhibitors cobimetinib (targeting MEK) and ulixertinib (targeting ERK) for targeted intervention at different levels of the RAS-MAPK cascade.

Cobimetinib (GDC-0973, XL518) is a highly selective and potent inhibitor of mitogen-activated protein kinases MEK1 and MEK2—key kinases within the RAS-RAF-MEK-ERK signaling cascade. As an allosteric inhibitor of the ATP-binding domain of MEK1/2, cobimetinib effectively suppresses the phosphorylation and activation of its downstream effector, ERK1/2 [50]. Due to its high selectivity, cobimetinib has minimal impact on other kinases, resulting in a relatively predictable toxicity profile compared to broader-spectrum inhibitors. Preclinical studies have demonstrated that cobimetinib effectively inhibits the proliferation of cell lines harboring activating mutations in *BRAF* (V600E), as well as in *KRAS* and *NRAS* [51]. In vivo, using xenograft models of melanoma and colorectal cancer, cobimetinib demonstrated significant tumor growth suppression that correlated with reduced levels of phospho-ERK in tumor tissue [50,52]. The combination of cobimetinib and venetoclax demonstrates maximal efficacy in patients with t(11;14) translocation due to synergistic suppression of BCL-2 and MCL-1 [48].

Ulixertinib (BVD-523) is a small molecule that functions as a highly selective, reversible, and ATP-competitive inhibitor of ERK1 and ERK2 (extracellular signal-regulated kinases)—the ultimate effectors of the highly conserved RAS-RAF-MEK-ERK signaling pathway. Unlike inhibitors targeting upstream kinases (BRAF, MEK), targeting ERK itself enables overcoming multiple resistance mechanisms that emerge due to pathway reactivation. Ulixertinib demonstrates potent anti-tumor activity in preclinical studies using cell lines and xenograft models with mutations in *BRAF*, *RAS*, and *MEK* genes, as well as in cases of acquired resistance to BRAF and MEK inhibitors [40,53]. Ulixertinib is currently being evaluated in phase I/II clinical trials for patients with advanced solid tumors exhibiting MAPK pathway activation [54].

The obtained data suggest that cobimetinib possesses an off-target effect involving P-gp inhibition. Functional analysis using the Rh123 efflux assay demonstrated that at cytotoxic concentrations (80% cell death), cobimetinib significantly suppressed P-gp activity, manifested by increased intracellular accumulation of the fluorescent dye up to 87.1% of the control level. This effect was not observed at non-toxic concentrations.

To verify direct interaction between cobimetinib and P-gp, molecular docking was performed. The results demonstrated high binding affinity scores, indicating stable interaction, with cobimetinib occupying a characteristic radial position within the molecule common to known P-gp inhibitors. Double-docking of cobimetinib and CFZ revealed that cobimetinib interacts with P-gp protein in a manner that prevents CFZ binding, as no molecular interactions between CFZ and P-gp were observed. Concurrently, it was established that 72 h incubation with cobimetinib at the IC_50_ concentration led not only to the expected suppression of the MAPK signaling pathway but also to significant downregulation of P-gp protein expression. Since we observed no P-gp inhibition with cobimetinib at non-toxic concentrations but detected cellular sensitization to CFZ in both cell lines, along with potent MAPK pathway suppression, we propose that cobimetinib sensitizes cells to CFZ primarily through MAPK cascade inhibition. However, at cytotoxic concentrations, cobimetinib exhibits a dual mechanism of action: direct inhibition of P-gp function combined with MAPK signaling suppression and subsequent downregulation of P-gp expression.

This study demonstrates that ulixertinib, similar to cobimetinib, exhibits off-target activity against P-gp. Functional analysis using the Rh123 efflux assay revealed dose-dependent P-gp inhibition: at a low-toxicity concentration, intracellular dye accumulation increased to 87.6%, while at 20 μM (corresponding to 20% cell death), this parameter reached 92.5%. These findings were corroborated by molecular docking results showing stable interaction between ulixertinib and P-gp. Importantly, unlike cobimetinib, ulixertinib demonstrated less pronounced MAPK pathway suppression activity in AMO-1/CFZ cells. However, 72 h incubation at the IC_50_ concentration resulted in reduced P-gp expression. These results suggest that the primary mechanism underlying ulixertinib-mediated sensitization of AMO-1/CFZ cells to CFZ is direct inhibition of P-gp functional activity.

Our findings are consistent with existing literature. Specifically, Ji et al. (2018) [55] demonstrated that ulixertinib effectively overcomes MDR mediated by ABCB1 and ABCG2 through direct inhibition of their transport function and increased intracellular accumulation of chemotherapeutic drugs. The concordance of mechanisms of action identified in independent studies underscores the universal nature of ulixertinib’s interaction with ABC transporters. Thus, we have demonstrated that inhibitors of key MAPK cascade components—ERK (ulixertinib) and MEK (cobimetinib) possess sensitizing properties that effectively restore CFZ sensitivity in P-gp-positive cells.

It should be noted that this study utilized the AMO-1 cell line, which carries a specific *RAS* gene mutation. The RAS-MAPK pathway is known to be the most frequently mutated pathway in MM, with K-RAS being a common isoform across human tumors. *KRAS* and *NRAS* mutations occur in approximately 40–50% of MM patients, particularly at the relapse/refractory stage. These mutations lead to constitutive activation of downstream signaling, rendering cells less dependent on external stimuli and more resistant to therapy [44,56]. While our findings reflect mechanisms within a specific cellular context—the AMO-1 line with a *RAS* mutation—this genetic background is highly relevant and commonly observed in MM patients. Nevertheless, further validation in additional models is warranted. A further consideration is the use of a single cell line, which may limit the generalizability of our findings. This is particularly relevant in MM, where *ABCB1* expression is known to be highly heterogeneous. For instance, Besse et al., in a large cohort that included patients resistant to bortezomib (BTZ-R, n = 33) and carfilzomib (CFZ-R, n = 29), newly diagnosed patients (NDMM, n = 1309), and circulating malignant plasma cells (n = 44), demonstrated that while *ABCB1* expression was highly variable, it was significantly upregulated in the circulating plasma cells of resistant patients [6]. Building on these data, it can be hypothesized that P-gp overexpression in MM cells may be significant for disease relapse. Furthermore, the clinical relevance of *ABCB1* is underscored by the discovery of an *APE1/YB-1/ABCB1* gene signature associated with adverse outcomes, a finding supported by significant co-expression of *YB-1* and *ABCB1* in a cohort of 22 MM patients [57]. This suggests that *ABCB1* upregulation, while observed, is not a universal event in MM. Therefore, expanding our experimental models, particularly through the inclusion of patient-derived data, remains essential to validate and contextualize our results.

## 4. Materials and Methods

### 4.1. Cells

The human multiple myeloma cell lines AMO-1 (obtained from the Leibniz Institute DSMZ) [58] and resistant to CFZ AMO-1/CFZ were cultured in RPMI 1640 medium (PanEco, Moscow, Russia) supplemented with 20% fetal bovine serum (FBS) (Biowest, Nuaille, France) and with 50 µg/mL streptomycin and 50 U/mL penicillin (PanEco, Russia). Cells were maintained at 37 °C in a humidified atmosphere containing 5% CO_2_. The AMO-1/CFZ cell line was established by cultivation of AMO-1 cells with progressively increasing concentrations of CFZ.

### 4.2. Drugs and Reagents

The following compounds were used in this study: Carfilzomib (Cayman Chemical, Ann Arbor, MI, USA, № 17554, CAS 868540-17-4), Cobimetinib (Bidepharm, Shanghai, China, № BD303547, CAS 934660-93-2), Ulixertinib (Bidepharm, Shanghai, China, № BD517941, CAS 1956366-10-1), Elacridar (Cayman Chemical, Ann Arbor, MI, USA, № 18128, CAS 143664-11-3). Stock solutions of the carfilzomib, cobimetinib and ulixertinib were prepared in dimethyl sulfoxide (DMSO; PanReac Appli-Chem, Monza, Italy, CAS. 67-68-5) at a concentration of 5 mM.

### 4.3. Flow Cytometry and Cell Sorting

AMO-1 and AMO-1/CFZ cells were stained with a FITC-conjugated mouse monoclonal antibody against human P-glycoprotein (clone 17F9; BD Biosciences, San Jose, CA, USA) for 30 min in the dark without prior fixation, then washed twice with phosphate-buffered saline. Analysis was performed on a FACSAria SORP instrument (BD Biosciences, USA) equipped with a 488 nm laser and 525/50 nm bandpass emission filter. Stained AMO-1/CFZ cells were sorted into two distinct subpopulations, P-gp LOW and P-gp HIGH (Figure 2), using the same instrument. Purity of the isolated subpopulations was confirmed by immediate post-sort analysis. Four independent experiments were performed. Data were acquired using Diva 8.1.1 software and analyzed using FlowJo v10 software (BD Biosciences, USA).

### 4.4. RNA Sequencing

Total RNA was extracted from cell lines (n = 4 per group across three experimental groups) using TRIzol™ Reagent (Thermo Fisher Scientific, Waltham, MA, USA) according to the manufacturer’s protocol. All RNA samples met the following quality control criteria: RNA Integrity Number (RIN) ≥ 7, as determined by capillary electrophoresis (Agilent 2100 Bioanalyzer), a concentration ≥ 40 ng/μL, and a total mass ≥ 1 μg. Sequencing libraries were prepared from qualified RNA samples and sequenced on a SURFSeq 5000 platform (Illumina, San Diego, CA, USA) to generate at least 40 million paired-end reads per library. The RNA sequencing data are available in the GEO repository under the accession number GSE310880. Public access to the data enabled on November 23, 2025.

### 4.5. Bioinformatic Analysis

Analysis of raw sequencing reads was performed using a Linux command-line pipeline. Raw read quality control (QC) was performed with FastQC [59] followed by alignment to the GRCh38 reference genome (GENCODE v42 annotation) using STAR [60]. Post-alignment quality and feature-assignment metrics were computed with QoRTs [61]. Gene-level counts were generated with featureCounts [62] against exon annotations. The pipeline-wide QC report was compiled with MultiQC [63]. Downstream analysis of gene-level counts was conducted in R software (version 4.2.1). Lowly expressed genes were filtered with the filterByExpr function from edgeR package (version 3.40.0) [64]. Differential gene expression analysis was performed with DESeq2 (version 1.38.0) [65], using the Wald test and controlling the false discovery rate using Benjamini–Hochberg adjustment (FDR). Variance-stabilized counts (VST) were used for exploratory analysis and visualization (PCA, clustering, heatmaps). Differentially expressed genes (DEGs) were visualized with the following tools: pheatmap (RRID:SCR_016418), EnhancedVolcano (RRID:SCR_018931), ggplot2 (RRID:SCR_014601), and the web tool Phantasus [66]. Gene set enrichment analysis was performed with fgsea [67] on genes ranked by signed test statistic, using Hallmark, KEGG, GO, and Reactome pathway collections.

### 4.6. Cytotoxicity

For the cytotoxicity assessment the cells were seeded into 96-well cell plates (SPL Lifesciences, Pyeongtaek, South Korea) at a density of 25 × 10^3^ cells in 135 μL of culture medium. Then 15 μL per well of increasing concentrations of compounds were added right after on the same day to final concentrations 0–1000 nM for CFZ and 0–100 mM for cobimetinib, ulixertinib. For experiments with ELA, the cells were seeded into 96-well cell plates at a density of 35 × 10^3^ cells in 120 μL of culture medium and treated with CFZ alone (15 μL CFZ + 15 μL medium without FBS) or a combination of CFZ 15 μL + ELA 15 μL. Non-toxic concentrations of ELA were used: 3 μM for AMO-1 cells and 12 μM for AMO-1/CFZ cells. In subsequent experiments, CFZ was administered in combination with low-toxic concentrations of the drugs ulixertinib (10 μM) or cobimetinib (1 μM). For this, the cells were seeded into 96-well cell plates at a density of 25 × 10^3^ cells in 120 μL of culture medium and treated with 30 μL per well of solutions: CFZ 15 μL + inhibitor (15 μL) or CFZ (15 μL) + medium without FBS (15 μL) right after on the same day. Wells treated with medium, which does not contain FBS, were used as control wells. Wells treated with 15 μL of inhibitor and 15 μL of medium without FBS were used to verify the viability of cells at low-toxic concentration of the inhibitor (cobimetinib or ulixertinib).

Then the cells were incubated for 72 h (or 24 h for tests with CFZ in combination with ELA) at 37 °C in the atmosphere with 5% CO_2_. Subsequently, 20 μL of MTT (5 mg/mL) (PanEco, Moscow, Russia) was added to each well. After incubation for 2 h, the medium was removed, and the formazan crystals were dissolved in 60 μL of DMSO (COMPONENT-REAKTIV, Moscow, Russia). The optical density was measured using the Multiskan FC spectrophotometer (Thermo Fisher Scientific, Waltham, MA, USA) at a wavelength of 540 nm. For the determination of half-maximal inhibitory concentration (IC_50_) and low-toxic concentrations, Skanlt RE 6.1.1 software was used, and dose–response curves were plotted. At least four wells were used for each concentration of substances or control, and MTT tests were performed in four experiments.

### 4.7. Quantitative Real-Time Polymerase Chain Reaction (RT-qPCR)

Total RNA from cells was isolated using ExtractRNA reagent (Evrogen, Moscow, Russia) according to the manufacturer’s protocol [68]. RNA integrity was confirmed by 1% (*w*/*v*) agarose gel electrophoresis. Complementary DNA (cDNA) was synthesized using RevertAid Reverse Transcriptase (RT), a recombinant M-MuLV RT (Thermo Fisher Scientific, Waltham, MA, USA) and 6 μL of total RNA. The reverse transcription reaction (final volume: 20 μL) included total RNA, H_2_O DEPC (Evrogen, Moscow, Russia), Random 6 primer (SYNTOL, Moscow, Russia), dNTP (SYNTOL, Russia), RNase inhibitor (SYNTOL, Russia) and RevertAid RT with Reaction buffer for RT. The mixture was incubated at 42 °C for 60 min, followed by enzyme inactivation at 70 °C for 10 min and dilution by 10 times in H_2_O. RT-qPCR was performed on a CFX Connect Real-Time System (BIO-RAD, Hercules, CA, USA) using Bio-Rad CFX Maestro1.1 software. The thermal cycling profile for gene expression included an initial denaturation step at 95 °C for 3 min, followed by 40 cycles of 10 s at 95 °C, 10 s at 60 °C, and 30 s at 72 °C. Next followed 5 s at 72 °C and 10 s at 95 °C as the final steps. The primer sequences used in this study are provided in Appendix A. Oligonucleotides were synthesized by SYNTOL, Russia. Human Large Ribosomal Protein (*RPLPO*) was used as the housekeeping gene for relative quantification. The RT-qPCR was carried out in triplicate for each gene of each sample.

### 4.8. Western Blot Analysis

Cells were harvested and lysed in RIPA buffer supplemented with a protease inhibitor cocktail (04693124001, Roche, Basel, Switzerland) and a phosphatase inhibitor cocktail (4906845001, Roche, Switzerland). Protein concentration was determined using the BCA Protein Assay Kit (71285-M, Millipore, Burlington, MA, USA). Protein samples were separated by SDS-PAGE on 7.5–10% gels and subsequently transferred electrophoretically onto nitrocellulose membranes. The membranes were blocked with 5% BSA for 1 h and then incubated overnight at 4 °C with primary antibodies. The following primary antibodies were used: Phospho-ERK Pathway Antibody Sampler Kit (9911, Cell Signaling Technology, Danvers, MA, USA), which includes antibodies against p-c-Raf (Ser338, dilution 1:1000), p-MEK1/2 (Ser217/221, dilution 1:1000), p-p44/42 MAPK (ERK1/2) (Thr202/Tyr204, dilution 1:2000), and p-p90RSK (Ser380, dilution 1:1000); P-glycoprotein (26528, Invitrogen, Carlsbad, CA, USA, dilution 1:1000); and β-Actin (47778, Santa Cruz, Dallas, TX, USA, dilution 1:1000). After incubation, the membranes were washed three times with TBST and incubated with corresponding horseradish peroxidase (HRP)-conjugated secondary antibodies for 1 h at room temperature. Protein bands were visualized using an ECL reagent (32209, Thermo Fisher Scientific, Waltham, MA, USA) and imaged with an ImageQuant Las4000 luminometer (GE Healthcare, Chicago, IL, USA). Band intensity quantified by densitometry using ImageJ software (Version 1.54g).

### 4.9. Evaluation of P-gp Functional Activity

To assess P-gp activity, AMO-1/CFZ cells were incubated for 20 min in a culture medium containing 2.0 µg/mL Rhodamine 123 (Rh123; Sigma-Aldrich, St. Louis, MO, USA). After incubation, the cells were washed twice with serum-free medium and divided into several aliquots (5 × 10^5^ cells per sample). One aliquot was incubated in a drug-free medium, and the others were incubated with the addition of the test compounds. The reference compound used was the P-gp inhibitor ELA. Incubation was carried out in RPMI-1640 culture medium without FBS at 37 °C for 50 min. Cell fluorescence was evaluated using a Becton Dickinson FACScan flow cytometer (BD Biosciences, San Jose, CA, USA). The results were analyzed using CellQuest software, version 3.3.

### 4.10. Molecular Docking

Molecular docking was performed to investigate interactions between ligands and P-gp. Pair molecular docking was performed using the Maestro software package (version 13.5.147) (Schrödinger, Inc., New York, NY, USA), and the 3D structure of the P-gp was applied by the open PDB database, with the PDBID identification number—6QEX. The structure of the human ABCB1 receptor protein was studied using electron microscopy with a resolution of 3.60 Å. This structure is described and implemented by the Institute of Molecular Biology and Biophysics (Switzerland) [69]. The first step of preparation was performed using the Protein Preparation Wizard panel. This process included the assignment of protonation states at pH 7.0, removal of water molecules and co-solvents, restoration of disulfide bonds, and optimization of missing side chains and loops. Then, a single minimization was performed using a force field OPLS3 (https://software.stanford.edu/software/opls3-force-field/, assessed on 4 September 2025). The putative active site was identified using the SiteMap tool (https://www.schrodinger.com/platform/products/sitemap/, assessed on 4 September 2025). The average rating of the active center for which the docking was performed is Site Score = 1.119; D score = 1.135. The docking location was allocated using the Receptor Grid Generation panel for localizing the best active center, with scaling factor = 1.0; partial charge cut off = 0.25, size box = 25 Å. Part of the ligands in sdf-3D format was downloaded from PubChem open database; when absent, a part of the ligands’ 3D form in PubChem was downloaded from the PDB.

The Maestro 11 platform standard protocol was used to quantify interactions between target proteins and ligands. Docking performed by standard protocol Glide docking SP Maestro, parameter scaling factor = 1.0; partial charge cut off = 0.25. Then, we assessed the position of the ligand in the active site of the receptor protein and analyzed the bonds using the Ligand Interaction panel. Double docking was carried out using a two-step algorithm—firstly, docking with the alleged inhibitor (elacridar/cobimetinib/ulixertinib) was carried out, then the inhibitor-embedded receptor protein was exported and re-docking with the substrate (CFZ) was applied.

### 4.11. Statistical Analyses

Statistical analyses were performed using GraphPad Prism 9.5.1 software. The results are presented as mean (M) ± standard deviation (SD). The Shapiro–Wilk test was performed to verify if data were normally distributed. For comparisons between the two groups, an unpaired Student’s *t*-test or Mann–Whitney test was applied. Multiple comparisons were performed using one-way ANOVA, followed by Tukey’s post-test, Dunnett’s multiple comparisons test, or unpaired Student’s *t*-test with Welch’s correction. Differences were considered significant when *p* < 0.05.

## 5. Conclusions

The obtained data suggest that the induction of P-gp expression in AMO-1 cells under carfilzomib treatment is mediated by activation of the MAPK signaling pathway. This makes targeted therapy of this pathway a promising strategy for overcoming CFZ resistance in MM cells. The ERK inhibitor ulixertinib and the MEK inhibitor cobimetinib have demonstrated potential as sensitizing agents, restoring the sensitivity of P-gp-positive cells to carfilzomib.

## Figures and Tables

**Figure 1 ijms-26-11448-f001:**
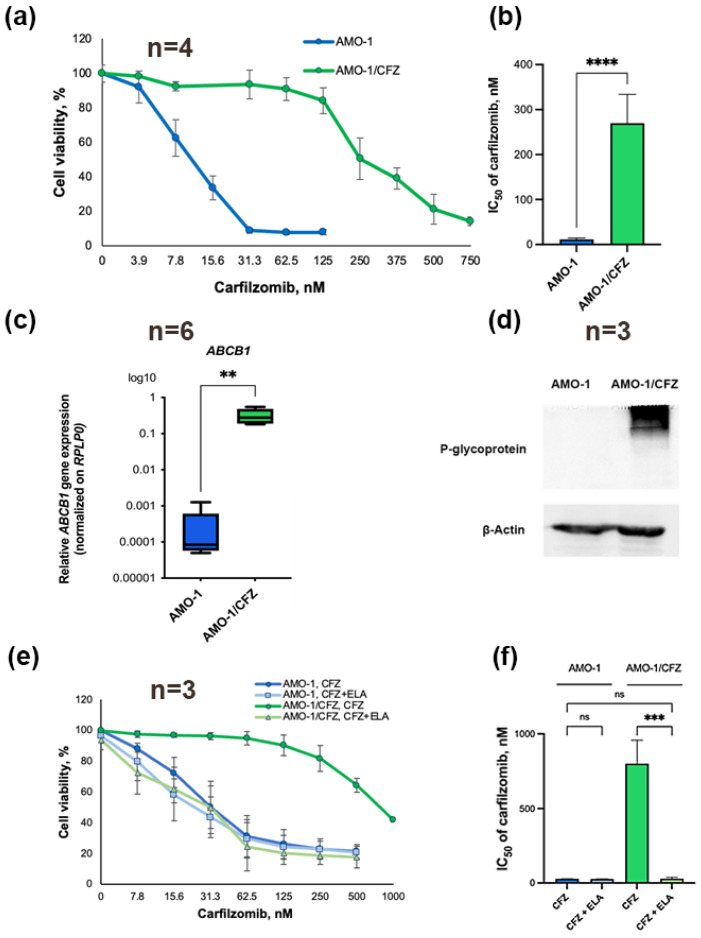
P-glycoprotein (P-gp) overexpression is the primary mechanism of carfilzomib (CFZ) resistance in AMO-1/CFZ cells. (**a**) Dose–response curves of AMO-1 and AMO-1/CFZ cells treated with CFZ for 72 h. Cell viability was measured by MTT assay; (**b**) CFZ IC_50_ values for AMO-1 and AMO-1/CFZ cells are shown. Data were analyzed using an unpaired *t*-test; (**c**) *ABCB1* mRNA levels in AMO-1 and AMO-1/CFZ cells were quantified by reverse transcription-quantitative PCR (RT-qPCR). Relative gene expression (normalized to *RPLP0*) was calculated via the 2^(–ΔCt) method. Data were analyzed using the Mann–Whitney U test; (**d**) Western blot analysis of P-gp expression in AMO-1 and AMO-1/CFZ cells (a representative blot from three independent experiments is shown). β-actin was used as a loading control to confirm equal protein loading (**e**) Dose–response curves for AMO-1 and AMO-1/CFZ cells treated with CFZ alone or a combination of CFZ and P-gp inhibitor elacridar (ELA) for 24 h, MTT assay. The following non-toxic concentrations of ELA were used: 3 μM for AMO-1 and 12 μM for AMO-1/CFZ cells; (**f**) CFZ IC_50_ values for AMO-1 and AMO-1/CFZ cells treated with CFZ alone or a combination of CFZ + ELA are shown. Data were analyzed using one-way ANOVA with Tukey’s post hoc test. The number of independent experiments (*n*) is indicated on the graphs. The values represent the mean ± SD; ns—no significant; **—*p* < 0.01, *** *p* < 0.001, **** *p* < 0.0001.

**Figure 2 ijms-26-11448-f002:**
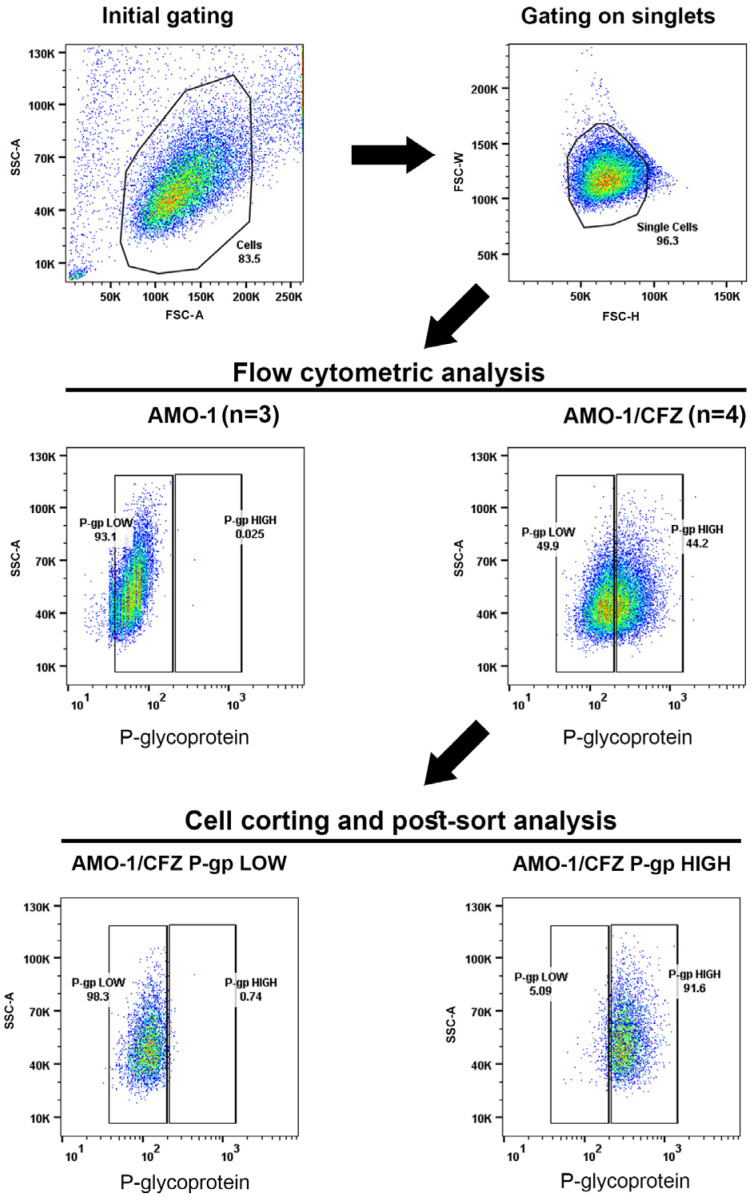
P-glycoprotein (P-gp) expression in AMO-1 and AMO-1/CFZ cells and cell sorting strategy. The gating strategy is shown on the upper panel. P-gp expression in AMO-1 and AMO-1/CFZ is shown in the middle panel. Post-sort analysis for AMO-1/CFZ P-gp LOW and P-gp HIGH subpopulation presented on the bottom panel. The results of one of the four experiments are presented. The number of independent experiments (*n*) is indicated in the middle panel.

**Figure 3 ijms-26-11448-f003:**
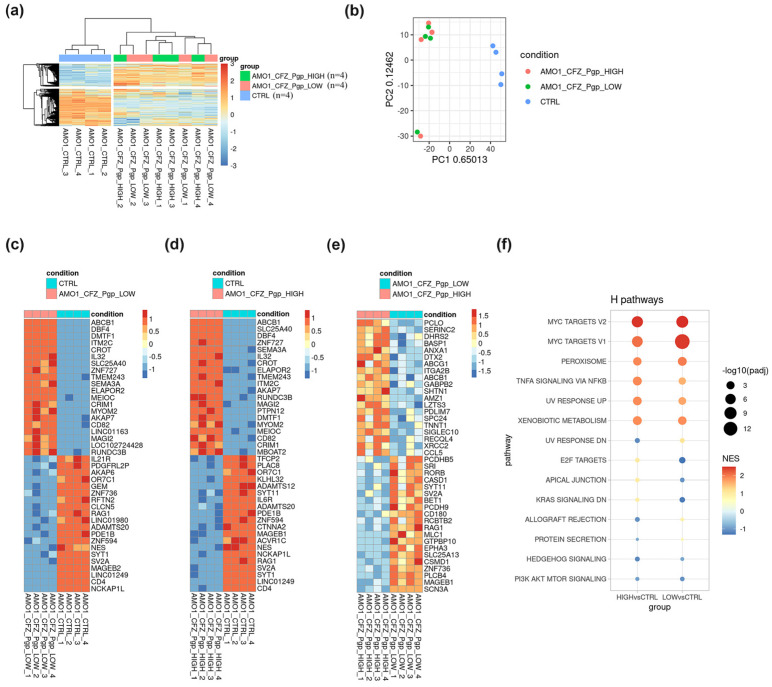
Results of RNA sequencing analysis of the three sample groups: AMO-1 (CTRL), AMO-1/CFZ_P-gp LOW, and AMO-1/CFZ_P-gp HIGH. (**a**) Hierarchical clustering of gene expression across the groups; (**b**) Principal component analysis (PCA) of the samples; (**c**) Heatmap of the top 20 most differentially expressed genes between the AMO-1 and AMO-1/CFZ_P-gp LOW groups; (**d**) Heatmap of the top 20 most differentially expressed genes between the AMO-1 and AMO-1/CFZ_P-gp HIGH groups; (**e**) Heatmap of the top 20 most differentially expressed genes between the AMO-1/CFZ_P-gp LOW and AMO-1/CFZ_P-gp HIGH groups; (**f**) The most significantly differentially enriched signaling pathways between the studied groups.

**Figure 4 ijms-26-11448-f004:**
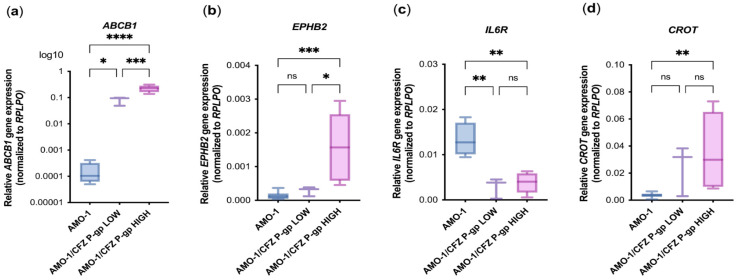
Verification of the sequencing results using RT-qPCR. Relative gene expression in AMO-1 cells (n = 4) and AMO-1/CFZ P-gp LOW (n = 3), AMO-1/CFZ P-gp HIGH (n = 4), separated by cell sorting: (**a**) *ABCB1* gene expression; (**b**) *EPHB2* gene expression; (**c**) *IL6R* gene expression; (**d**) *CROT* gene expression. Gene expression was normalized to the housekeeping gene *RPLPO*. The statistical ANOVA test with Tukey’s multiple comparisons post-test was used. ns—not significant, *—*p*< 0.05, **—*p* < 0.01, ***—*p* < 0.001, ****—*p* < 0.0001.

**Figure 5 ijms-26-11448-f005:**
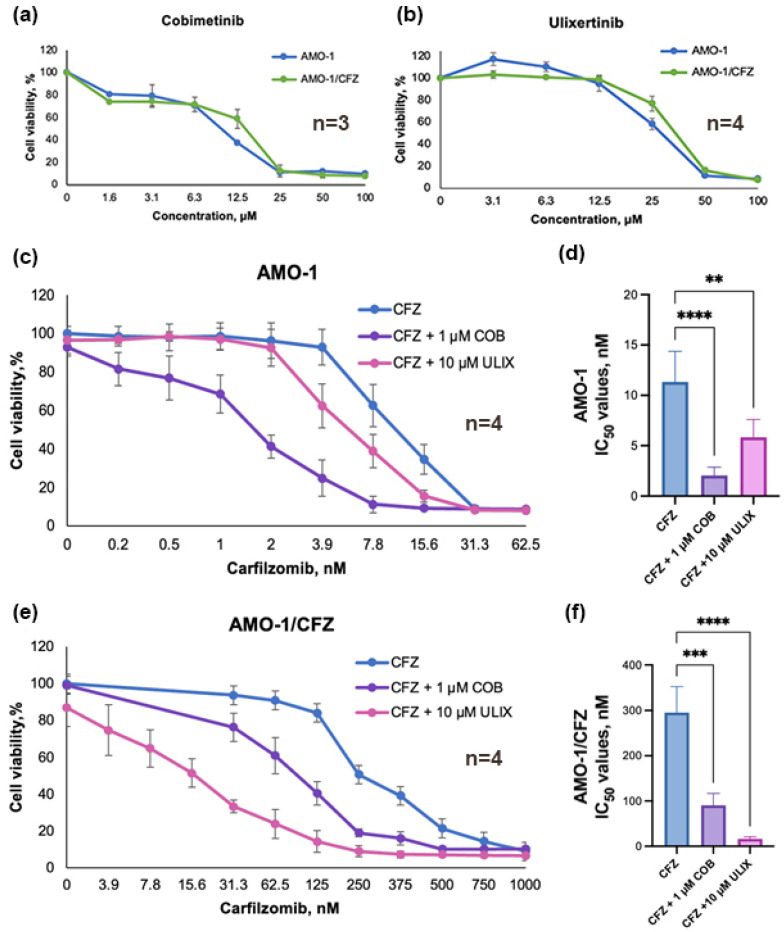
Dose–response curves for AMO-1 and AMO-1/CFZ cell lines treated with kinase inhibitors alone or in combination with CFZ. (**a**) MEK1/2 kinase inhibitor cobimetinib (COB); (**b**) ERK1/2 protein kinase inhibitor ulixertinib (ULIX); (**c**) dose–response curves for CFZ in combination with ULIX or COB in AMO-1 cells; (**d**) CFZ IC_50_ values for AMO-1 cells treated with CFZ alone or a combination CFZ + COB or ULIX; (**e**) dose–response curves for CFZ in combination with ULIX or COB in AMO-1/CFZ cells; (**f**) CFZ IC_50_ values for AMO-1/CFZ cells treated with CFZ alone or a combination CFZ + COB or ULIX. Point (0;100) represents experiments in which only low-toxic concentrations of COB or ULIX were added to the cells. The number of independent experiments (*n*) is indicated on the graphs. Statistical analysis was conducted using ANOVA with Dunnett’s multiple comparisons test. **—*p* < 0.01, ***—*p* < 0.001, ****—*p* < 0.0001.

**Figure 6 ijms-26-11448-f006:**
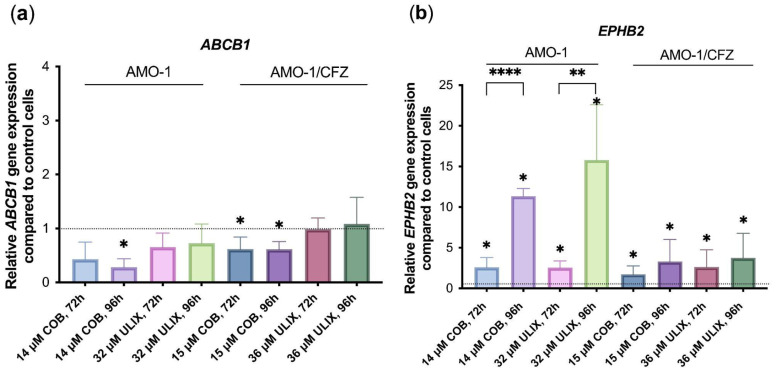
Relative (**a**) *ABCB1*, (**b**) *EPHB2* gene expression in AMO-1 and AMO-1/CFZ cells after exposure to IC_50_ of cobimetinib (COB) or IC_50_ of ulixertinib (ULIX) for 72 and 96 h. Gene expression was quantified by RT-qPCR and normalized to the housekeeping gene *RPLPO*. The values are presented as the ratio between cells treated with inhibitors and control cells from five experiments. The dotted lines denote the gene expression level in the control samples, defined as 1. Statistical analysis was conducted using ANOVA multiple comparisons test with unpaired Student’s *t*-test with Welch’s correction. *—*p* < 0.05, **—*p* < 0.01, ****—*p* < 0.0001.

**Figure 7 ijms-26-11448-f007:**
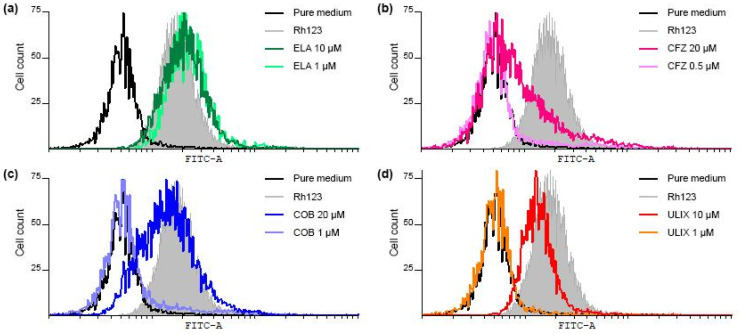
Flow cytometric analysis of P-gp function using Rhodamin 123 (Rh123) efflux assay in P-glycoprotein overexpressing AMO-1/CFZ cells. Following a 20 min incubation period with Rh123, the cells were further incubated for 40 min in either pure medium or medium containing (**a**) elacridar (ELA), (**b**) carfilzomib (CFZ), (**c**) cobimetinib (COB), or (**d**) ulixertinib (ULIX). Data from one representative experiment out of three are shown for each compound.

**Figure 8 ijms-26-11448-f008:**
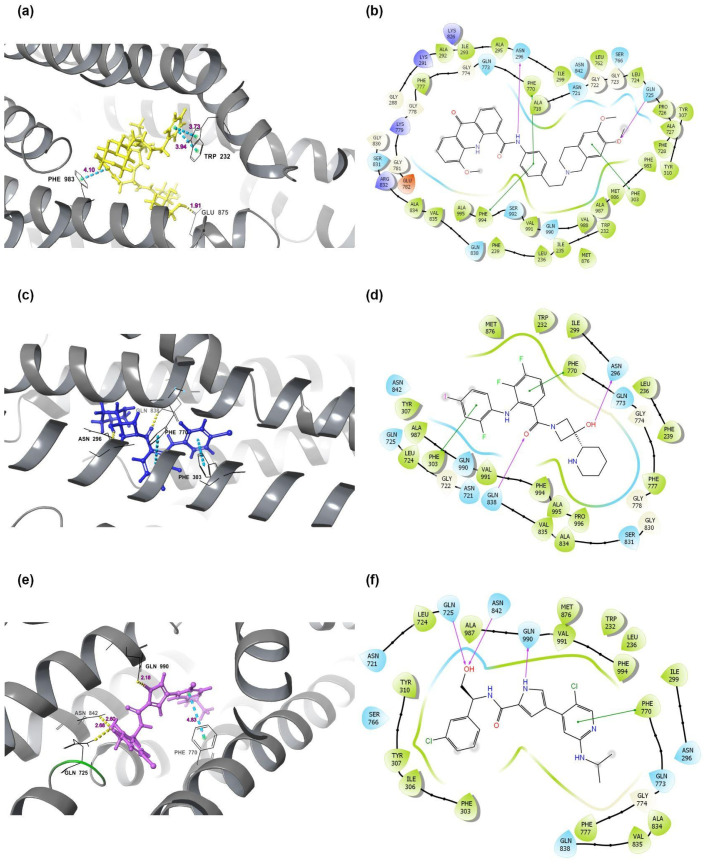
Molecular docking of different substances to P-glycoprotein (P-gp, PDB code: 6QEX). (**a**,**b**) Molecular docking of potential substrate Carfilzomib into to P-gp, which forms 1 hydrogen bond between Gln875 and OH- group in molecular Carfilzomib; 3 p-stacking bonds—between Phe983 and benzole ring as well as two ones between Trp232 and another benzole ring in CFR; (**c**,**d**) Molecular docking of potential inhibitor Cobimetinib into to P-gp, which forms 2 hydrogen bonds—between Asp296 and Gln838 and two OH-groups in Cobimetinib molecula; 2 p-stacking bonds—between Phe303, Phe770 and two benzole rings in 2-3-ftornitrobenzole and 2-ftor-4iodnitrobenzole; (**e**,**f**) Molecular docking of potential inhibitor Ulixertinib into to P-gp, which forms 3 hydrogen bonds—between Gln725, Asp842 and one OH group, Gln990 and pirrole NH-group; 1 p-stacking bond between Phe770 and chlorpiridine ring of the Ulixertinib molecula. 3D- (**a**,**c**,**e**) and 2D-structures (**b**,**d**,**f**) are shown.

**Figure 9 ijms-26-11448-f009:**
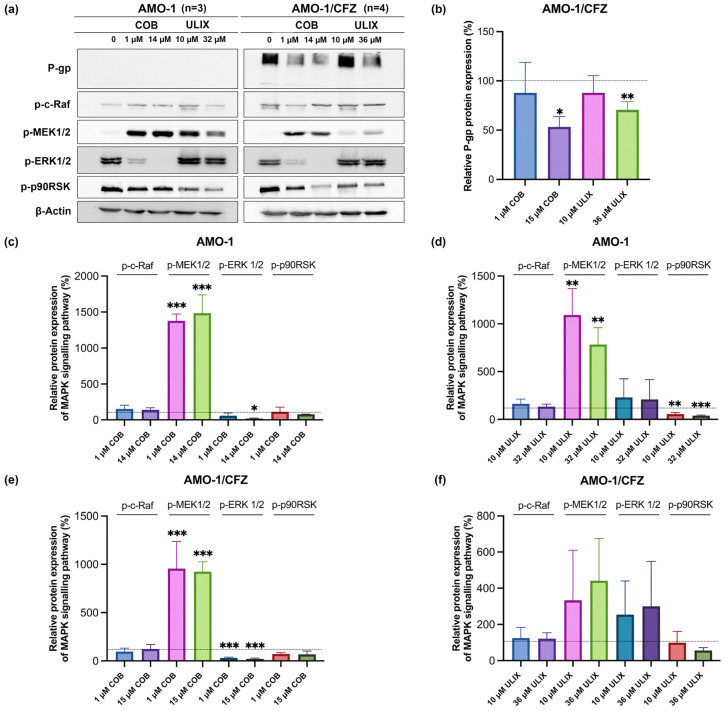
Effect of cobimetinib (COB) and ulixertinib (ULIX) on the MAPK signaling pathway and P-gp expression. (**a**) Representative Western blots of MAPK pathway proteins and P-gp in AMO-1 and AMO-1/CFZ cells; (**b**) Densitometry analysis of P-gp protein expression bands relative to control in AMO-1/CFZ cells; (**c**) Densitometry analysis of phosphorylated MAPK pathway proteins relative to control in AMO-1 cells treated with COB; (**d**) Densitometry analysis of phosphorylated MAPK pathway proteins relative to control in AMO-1 cells treated with ULIX; (**e**) Densitometry analysis of phosphorylated MAPK pathway proteins relative to control in AMO-1/CFZ cells treated with COB; (**f**) Densitometry analysis of phosphorylated MAPK pathway proteins relative to control in AMO-1/CFZ cells treated with ULIX. The dotted lines denote the protein expression level in the control samples, defined as 100%. The number of independent experiments (*n*) is indicated on the graphs. Results are presented as M ± SD. Statistical analysis was conducted using ANOVA with Dunnett’s multiple comparisons test. *—*p*< 0.05, **—*p* < 0.01, ***—*p* < 0.001.

**Table 1 ijms-26-11448-t001:** The IC_50_ (nM) values of CFZ in AMO-1 and AMO-1/CFZ cells treated with CFZ alone or in combination with low-toxic concentrations of cobimetinib or ulixertinib, and the sensitization of cell lines to carfilzomib (fold). Data are presented as the mean ± standard deviation (M ± SD).

	AMO-1	Sensitization (Fold)	AMO-1/CFZ	Sensitization (Fold)
CFZ	11.35 ± 3.03	-	295.52 ± 57.19	-
CFZ + cobimetinib	2.03 ± 0.85	5.6	90.6 ± 26.13	3.3
CFZ + ulixertinib	5.84 ± 1.77	1.9	16.52 ± 4.79	17.9

**Table 2 ijms-26-11448-t002:** In Silico molecular docking parameters for P-glycoprotein interaction with investigated compounds.

Substance Name	Substance Class in Relation to P-Glycoprotein	Docking Score	Position in the Molecule	Interaction
CARFILZOMIB (PDB: 3BV)	Substrate	−9627	Center	Pi-pi stacking (3), H-bond TRP232, GLU875, PHE983
Elacridar (PubChem: 119373)	Inhibitor	−9852	Radially	Pi-pi stacking (3), H-bond (2) ASN296, PHE303, GLN725, PHE770, PHE994
Cobimetinib (PubChem: 16222096)	Inhibitor	−8783	Radially	Pi-pi stacking (2), H-bond (2) PHE303, PHE770, ASN296, GLN838
Ulixertinib (PubChem: 11719003)	Inhibitor	−7698	Radially	Pi-pi stacking, H-bond (3) ASN842, GLN725, GLN990, PHE770
Elacridar + CARFILZOMIB		−10,725	Radially	Pi-pi stacking (2), H-bond (2) ASN296, PHE303, GLN725, PHE770
			Center	H-bond GLN946
Cobimetinib + CARFILZOMIB		−9739	Radially	Pi-pi stacking (2), H-bond (2) PHE303, GLN838, PHE770, ASN296
			Center	-
Ulixertinib + CARFILZOMIB		−10,044	Radially	Pi-pi stacking, H-bond (3) ASN842, GLN725, GLN990, PHE770
			Center	Pi-pi stacking (2), H-bond (2) PHE336, PHE983, GLU875, GLN195

## Data Availability

The data presented in this study are available in the article and Appendix A.

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
