# Peer review of "Targeting the MEK/ERK Pathway to Suppress P-Glycoprotein and Reverse Carfilzomib Resistance in Multiple Myeloma"

_ijms, 2025, doi:10.3390/ijms262311448_

Round 1

Reviewer 1 Report

Comments and Suggestions for Authors

In the manuscript, "Targeting the MEK/ERK Pathway to Suppress P-glycoprotein and Reverse Carfilzomib Resistance in Multiple Myeloma", the authors have demonstrated that the molecular mechanism involved in carfilzomib resistance in multiple myeloma is mediated by suppression of P-glycoprotein expression via the MEK/ERK pathway. Overall, the manuscript is well written and follows a logical sequence of experiments that support the central claim. A few concerns are as follows:

  1. The mechanism described is well known across many cancers, and the authors should clarify the novelty of their findings.
  2. The introduction section does not adequately cover existing literature and known information. The pathway is widely known for enhancing drug efflux in cancers by activating transcription factors such as AP-1 and c-ETS, which in turn upregulate ABCB1/P-glycoprotein expression. Those references have not been cited.
  3. The manuscript presents consistent results but it’s all based on a single cell-line, so we don’t really know how universal these findings are. The results for a single cell line maybe biased due to their genetic background and culture conditions, hence it is hard to say whether the conclusions are applicable widely without validation in other models. The manuscript would benefit from connecting the findings to more real-world contexts—such as patient-derived data, in vivo or ex vivo systems, and/or publicly available datasets. The addition of such studies would add to the biological significance and help in understanding the potential clinical impact.

  4. Figure 1B; 5D and 5F: The graph is difficult to understand. the authors need to represent that data in a different format. Why is the concentration plotted against the cell lines? what exactly are the data points that have been considered for the bar graph is not clear and difficult to guess too. 

Reviewer 2 Report

Comments and Suggestions for Authors

The manuscript "Targeting the MEK/ERK Pathway to Suppress P-glycoprotein and Reverse Carfilzomib Resistance in Multiple Myeloma", by Lidia A. Laletina and colleagues, aims to elucidate the signalling pathways underlying the emergence of P-glycoprotein (P-gp) overexpression in multiple myeloma (MM) cells, which contributes to resistance to the proteasome inhibitor carfilzomib (CFZ). Several aspects of the manuscript require clarification:

  • Typographical errors should be corrected, including the representation of 103  instead of “103”, and the proper formatting of chemical formulas and temperature units, such as CO₂, H₂O, and 37 °C.
  • Figure 2 is particularly confusing; the dot plots would benefit from arrows indicating the gating strategy to guide the reader through the experimental workflow.
  • All figures should specify the sample size used for each experiment.
  • Tables 1 and 2 should be reformatted following scientific conventions, displaying only the top and bottom borders along with a horizontal line to separate groups.
  • The concentrations of primary antibodies used in the Western blot experiments must also be explicitly stated.
  • While the references are generally accurate, inclusion of more recent literature from the last five years is recommended, such that 40–50 % of the citations reflect the most current advances in the field.

Author Response

We thank the reviewer for their thorough review and valuable suggestions. We have carefully addressed all the points raised, and the corresponding changes have been incorporated into the manuscript. Our point-by-point responses are detailed below.

Comment 1: Typographical errors should be corrected, including the representation of 10³ instead of “103”, and the proper formatting of chemical formulas and temperature units, such as CO₂, H₂O, and 37 °C.

Response 1: We thank the reviewer for pointing this out. The manuscript has been meticulously reviewed, and all typographical errors, including the formatting of superscripts, chemical formulas, and temperature units, have been corrected.

Comment 2: Figure 2 is particularly confusing; the dot plots would benefit from arrows indicating the gating strategy to guide the reader through the experimental workflow.

Response 2: We agree with this suggestion. Figure 2 has been revised to include arrows and annotations that now provide a clearer, step-by-step visual guide to the gating strategy, making the experimental workflow easier to follow.

Comment 3: All figures should specify the sample size used for each experiment.

Response 3: The sample size (n) for each experiment is now indicated directly in the figures. In cases where including this information directly on the graph would significantly complicate the visual presentation, or where representative data from a single experiment are shown, the sample size is explicitly stated in the corresponding figure legend.

Comment 4: Tables 1 and 2 should be reformatted following scientific conventions, displaying only the top and bottom borders along with a horizontal line to separate groups.

Response 4: We have reformatted Tables 1 and 2 in accordance with standard scientific conventions, as suggested.

Comment 5: The concentrations of primary antibodies used in the Western blot experiments must also be explicitly stated.

Response 5: The dilutions of all primary antibodies used in the Western blot analyses have now been explicitly stated in the 'Materials and Methods' section.

Comment 6: While the references are generally accurate, inclusion of more recent literature from the last five years is recommended, such that 40–50 % of the citations reflect the most current advances in the field.

Response 6: We thank the reviewer for this recommendation. We have updated the reference list by incorporating recent literature from the last five years. This includes replacing several older citations with more current ones in the results and discussion sections and significantly expanding the introduction with the latest findings in the field. Approximately 45% of the citations now reflect the most recent advances.

Round 2

Reviewer 1 Report

Comments and Suggestions for Authors

All concerns raised in the previous round of review have been adequately addressed.